# Past and Future Alcohol-Attributable Mortality in Europe

**DOI:** 10.3390/ijerph17239024

**Published:** 2020-12-03

**Authors:** Fanny Janssen, Shady El Gewily, Anastasios Bardoutsos, Sergi Trias-Llimós

**Affiliations:** 1Netherlands Interdisciplinary Demographic Institute, KNAW/University of Groningen, Lange Houtstraat 19, 2511 CV The Hague, The Netherlands; 2Population Research Centre, Faculty of Spatial Sciences, University of Groningen, P.O. Box 800, 9700 AV Groningen, The Netherlands; shadyelgewily@gmail.com (S.E.G.); a.bardoutsos@rug.nl (A.B.); 3Centre d’Estudis Demogràfics, Centres de Recerca de Catalunya (CERCA), Carrer de Ca n’Altayó, Buildings E2, Autonomous University of Barcelona, 08193 Bellaterra, Spain; strias@ced.uab.es; 4Department of Non-Communicable Disease Epidemiology, Faculty of Epidemiology and Population Health, London School of Hygiene & Tropical Medicine, Keppel St, London WC1E 7HT, UK

**Keywords:** alcohol, Europe, future, mortality, alcohol-attributable mortality, time trends

## Abstract

Although alcohol consumption is an important public health issue in Europe, estimates of future alcohol-attributable mortality for European countries are rare, and only apply to the short-term future. We project (age-specific) alcohol-attributable mortality up to 2060 in 26 European countries, after a careful assessment of past trends. For this purpose we used population-level country-, sex-, age- (20–84) and year-specific (1990–2016) alcohol-attributable mortality fractions (AAMF) from the Global Burden of Disease (GBD) study, which we adjusted at older ages. To these data we apply an advanced age-period-cohort projection methodology, that avoids unrealistic future differences and crossovers between sexes and countries. We project that in the future, AAMF levels will decline in all countries, and will converge across countries and sexes. For 2060, projected AAMF are, on average, 5.1% among men and 1.4% among women, whereas in 2016 these levels were 10.1% and 3.3%, respectively. For men, AAMF is projected to be higher in Eastern and South-western Europe than in North-western Europe. All in all, the share of mortality due to alcohol is projected to eventually decline in all 26 European countries. Achieving these projected declines will, however, require strong ongoing public health action, particularly for selected Eastern and North-western European countries.

## 1. Introduction

Alcohol consumption is an important public health challenge in Europe, as it substantially increases the risk of contracting many different diseases, including liver cirrhosis, alcohol use disorders, specific cancers, cardiovascular diseases, and infectious diseases, and the risk of sustaining (road/transport) injuries [1,2]. As alcohol use can have both acute and chronic health effects, it is a leading risk factor not just for disability, but for death [3,4]. Consequently, alcohol use has a large impact on (healthy) life expectancy [5,6]. Worldwide, Europe is the region with the highest levels of alcohol consumption: in 2016, an estimated 9.8 litres of pure alcohol were consumed per capita in Europe, compared to 6.4 litres worldwide [7]. Without alcohol-attributable mortality, national life expectancy levels in Europe would be, on average, 1.8 years higher among men and 0.5 years higher among women [8]. Thus, having a clear overview of past trends in alcohol-attributable mortality, and understanding how alcohol-attributable mortality is likely to further develop in the future, is highly relevant for society, and for health policy-makers in particular.

The few previous studies on alcohol-attributable mortality that employed both a comparative and a temporal approach revealed important country differences within Europe [4,5,6,9,10,11,12]. In general, Eastern European countries experience high levels of alcohol attributable mortality, particularly among men, with strong increases (mainly in the 1990s), followed by recent declines. Southern European countries, however, generally experienced declining trends over the last 30–40 years, resulting in lower levels. In Western and Northern European countries trends in alcohol-attributable mortality over recent decades are less clear and tend to stagnate with periods of modest increase as well as periods with modest decrease. Studies for selected European countries demonstrated the importance of the birth cohort dimension in describing and explaining past trends in (age-specific) alcohol-attributable mortality [9,10], in line with the previously observed cohort effects in alcohol use [13,14]. These findings suggest that individuals born in the same year tend to adopt similar drinking behaviours during young adulthood, which affects their subsequent drinking patterns and alcohol-related health problems [9,10].

The few previous studies that forecasted alcohol-attributable mortality employed methodologies that mainly provided all-age estimates for the short-term future. Sheron et al. (2011, 2012) projected all-age alcohol-attributable liver death rates and numbers in the United Kingdom up to 2030/2036 by linearly extrapolating the observed increase over the last 8–10 years, and by applying as alternative scenarios the past declining trends from either France, Italy, or the European Union [15,16]. Pruckner et al. (2019) linearly projected declines between 1979 and 2015 in age-standardised death rates for selected alcohol-related causes of death up until 2030 for 29 European countries [17]. Rosén and Haglund (2019) obtained, as well, an estimate of overall alcohol-related mortality for the short-term future only (for 2025), for Sweden, while considering the cohort dimension [18]. They applied age-period-cohort modelling to mortality from four main alcohol-related causes from 1969 through 2015. Their assumption of unchanged effects of cohort and age is, however, questionable. Trias-Llimós et al. (2020) employed a more advanced age-period-cohort methodology to project age-specific mortality from four main alcohol-related causes plus liver cirrhosis in France up to 2050, and demonstrated the relevance of cohort effects in all-age and age-specific projections [19].

These previous studies estimated and projected alcohol-attributable mortality based on causes of death, while either including all deaths from causes that are only partly related to alcohol (e.g., external causes of death) [17], or excluding these deaths [18,19]. However, neither approach fully reflects the whole impact of alcohol on mortality [20].

Our objective is twofold: first, to provide an overview of current levels and past trends in alcohol-attributable mortality in 26 European countries (1990–2016) using estimates that better reflect alcohol-attributable mortality; and, second, to obtain—for the first time—estimates of age-specific and age-standardised alcohol-attributable mortality for the long-term future in 26 European countries, while accounting for the cohort dimension and avoiding unrealistic future differences and crossovers between sexes and countries.

## 2. Material and Methods

### 2.1. Setting

We studied past trends in age-specific and age-standardised alcohol-attributable mortality fractions, and projected these trends into the future for the national populations aged 20–84, by sex, in 26 European countries (see Appendix B
Table A1). These 26 European countries represent all of the European countries for which there are both Global Burden of Disease (GBD) study data on alcohol-attributable mortality (1975–2016) and high-quality all-cause mortality data from the Human Mortality Database (HMD) since at least 1990 and up to at least 2013, except Estonia, Latvia and Slovakia, which we excluded from our analysis, because of their unrealistic alcohol-attributable mortality rates at the (very) old ages (see Appendix A for more details).

We studied past trends over the period 1990 up to 2016, or the latest available year (LAY). Consequently, we could study the cohorts born between 1906 (year 1990 minus age 84) and 1996 (year 2016 minus age 20).

In presenting our results, we distinguished between Eastern and Western European countries, but also further categorised the Western European countries into two main groups based on their clearly different past trends in age-standardised alcohol-attributable mortality fractions (AAMFs).See Figure 1 and its description as part of the Results section.

### 2.2. Alcohol-Attributable Mortality Estimates

We derived at population level estimates of alcohol-attributable mortality that reflect alcohol-attributable mortality more accurately than previous estimates. That is, we used the GBD 2017 estimates of alcohol-attributable mortality. Unlike previous estimates generated by an underlying cause-of-death approach, these estimates include both estimates of mortality from causes of death wholly attributable to alcohol and estimates of alcohol-attributable mortality from causes of death partly attributable to alcohol (see Appendix B
Box 0). However, in line with previous criticisms of the GBD estimates at the oldest ages, we adapted the alcohol-attributable mortality estimates from age 60 onwards.

More specifically, we obtained estimated population-level alcohol-attributable mortality rates by sex, five-year age group (20–24 to 80–84), single calendar year (1990–2016) and country from the GBD study 2017 [21]. By applying an attributable fraction (AF) approach, the GBD 2017 study estimated for each country the deaths that can be attributed to alcohol by sex and age [22]. To produce these estimates, the GBD used two types of data in addition to cause-specific mortality data: (i) alcohol consumption data, which they mainly obtained from a wide range of national representative health surveys; and (ii) information on the health risks of alcohol consumption (cause-specific relative risks at different levels of drinking) stemming from a systematic literature review [22]. By dividing the age- and sex-specific alcohol-attributable death numbers by the age- and sex-specific population numbers, the alcohol-attributable mortality rates were obtained.

The GBD estimates of alcohol-attributable mortality for the highest ages (65+) are, however, considered implausible, as they are either very high or negative (e.g., [20,23]). Among the explanations for these potential inaccuracies are that the estimation technique is highly dependent on the limited information on alcohol use at these ages; age-specific relative risks (RRs) of dying are lacking at these ages; and, more generally evidence regarding the impact of alcohol on health at those ages is lacking (e.g., [20,23]).

Therefore we adjusted the GBD estimates of alcohol-attributable mortality for ages 65+ using the more realistic age pattern for the highest ages [20]—but not the level—of causes of death wholly attributable to alcohol, using data from the World Health Organization (WHO) Mortality Database [24] and the Human Cause of Death Database [25]. For more details, see Appendix A.

### 2.3. Outcome Measure

Our outcome measure is the alcohol-attributable mortality fraction (AAMF), which can be interpreted as the share of deaths in a population that is due to alcohol. More generally, the population-attributable fraction (in our case, the AAMF) represents the proportion by which the outcome (here: all-cause mortality) would be reduced in a given year and in a given population if the exposure to a risk factor (here: alcohol consumption) was reduced to an ideal exposure scenario [22].

We obtained the AAMFs by dividing the adjusted age- and sex-specific alcohol-attributable mortality rates by the respective age- and sex-specific all-cause mortality rates from the HMD [26]. We, then, applied Loess smoothing to convert the AAMFs by five-year age groups into AAMFs by single year of age (AAMF_x,t_), to be used as input in our projections.

To allow for comparison over time, we estimated age-standardised AAMFs using the country- and sex-specific age distribution of deaths in 2010 from the HMD [26]. For the past trends (Figure 1) and current levels (Table 1—first column) we used the unsmoothed AAMFs as input for the age-standardised AAMFs.

### 2.4. Our Novel Approach to Project Alcohol-Attributable Mortality into the Future

We employed a novel projection approach that, unlike previously applied alcohol-attributable mortality projection methodologies which mainly provided all-age estimates for the short-term future [15,16,17,18], is able to provide realistic estimates of age-specific and age-standardised alcohol-attributable mortality for the long-term future.

That is, by employing age-period-cohort modelling, we took into account the importance of the cohort dimension, as well as the age and period dimensions, in determining past trends in alcohol-attributable mortality. In addition, we avoided unrealistic future crossovers and divergence in age-standardised AAMF between country groups and sexes, which can easily occur when extrapolating, for single countries, their different past trends into the long-term future. More specifically, by implementing assumed country group- and sex-specific lower limits of future age-standardised AAMF we ensured that the projected AAMF levels for men remained higher than those projected for women, who have, historically, always had lower AAMF levels. Similarly, based on past observations, we considered it unlikely that the higher current age-standardised AAMF values in Eastern European countries for men will become lower in the future than those in Western European countries. Moreover, we avoided unrealistically large future differences between countries in age-standardised AAMF levels by assuming that the current increases in AAMF observed for selected countries will eventually turn into declines. This assumption was motivated by the observation of a wave-shaped pattern of an increase followed by a decline in AAMF in a number of European countries (see Figure 1); by the large reductions in alcohol consumption that have recently occurred, particularly in Eastern Europe [27], by the decrease in alcohol use among the youth [14]; by the recent implementation of strong alcohol prevention policies in European countries [27]; and by evidence that prevention policies have the power to bend increases into declines [27].

We present our results up until 2060, which is well beyond the current projections (see Introduction), in order to facilitate mortality projections for almost the whole currently living adult population.

### 2.5. Methods

For more details, see Appendix A.

We projected the sex- and country-specific *AAMF_x,t_* up to 2060 by employing an advanced age-period-cohort (APC) projection methodology. We utilized the APC modelling approach by Clayton and Schifflers [28], which decomposes mortality into the shared linear trend between period and cohort (which is referred to as drift), a non-linear period effect, and a non-linear cohort effect. To simplify the interpretation and the projection, we clubbed the drift with the non-linear period effect using the Cairns et al. approach [29].

In applying the APC model to the *AAMF_x,t_*, we used a generalised logit as the link function. The logit transformation ensured projected *AAMFs* between zero and one, and enabled us to project (eventually) declining *AAMF* for countries with currently increasing *AAMF*. The generalisation enabled the implementation of the more restricted lower limits.

The model we applied is:(1)$$logit\left(\frac{AAMF_{x,t}-LB_{x}}{UB_{x}-LB_{x}}\right)={\tilde{\alpha }}_{x}+{\tilde{\kappa }}_{t}+{\tilde{\gamma }}_{t-x}$$
where *LB_x_* represents the time-constant but population-dependent age-specific lower bounds; *UB_x_* represents the age-specific upper bounds, which equal one; and $\alpha_{x}$, $\kappa_{t}$, and $\gamma_{t-x}$ capture the age pattern, the overall time trend, and the cohort deviations from the overall trend, respectively.

To obtain the age-specific lower bounds, we assumed different lower limits of age-standardised *AAMF* for different population groups (see Appendix A) in line with our general approach, and applied to these lower limits the population-specific age pattern observed in 2016/LAY.

For the projection of the period ($\kappa_{t}$) and cohort ($\gamma_{t-x}$) parameters—which we derived from the application of the aforementioned model—we employed different strategies according to their past trends (see Appendix A). The period parameter was projected by a quadratic curve with correlated errors for populations with predominantly increasing $\kappa_{t}$ trends; and, for populations with predominantly declining trends, by extrapolation of the (recent) decline by the best-fitting auto-regressive integrated moving average (ARIMA) model with drift, subject to some restrictions. When the decline in *k_t_* was followed by a recent increase, we projected a stable $\kappa_{t}$ trend. After omitting the outer three, five, or seven cohorts dependent on a statistical significance test, the recent trends for the cohort parameters were also projected by the best-fitting ARIMA model, subject to some restrictions. In the few cases in which this would lead to an increase, we projected a stable trend.

By performing 50,000 simulations, we obtained projected age-specific and age-standardised AAMF up to 2060, and their 95% projection intervals, by country and sex.

## 3. Results

In Europe in 2016/LAY, the age-standardised AAMFs (20–84) (= share of deaths due to alcohol) were high among Eastern European men (14%), particularly in Belarus (18%), and substantially lower among Western European men (8%), particularly in Norway and Iceland (5%) (Table 1).

The age-standardised AAMF were substantially lower among European women (3.3%) than European men (10.1%) (Table 1). Among women, the age-standardised AAMF ranged from 1% in Greece to over 5% in Luxembourg, and differences in AAMF levels between Eastern and Western Europe are small.

There were substantial differences between European countries in the trends over time (1990–2016) in AAMF (Figure 1). In the South-Western European countries of Austria, France, Germany, Greece, Italy, Portugal, Spain and Switzerland, AAMF diminished over the 1990–2016 period, albeit with considerable decelerations in the decline, and even periods of stagnation, particularly among men. For men and women in the North-Western and Eastern European countries, we observed either an increase followed by a decline (Denmark, Finland, Sweden, Ireland, Czech Republic, Hungary, Russia), an increase followed by stagnation (Belgium, Luxembourg (men), Norway, United Kingdom, Ukraine, Lithuania), or an ongoing increase (Iceland, Luxembourg (women), The Netherlands, Belarus, Poland).

The trends in the period parameter (*k_t_*) (Appendix A) largely resembled the trends in age-standardised AAMF, although differences also existed, indicating an additional effect of the cohort dimension. For example, for Austrian and German men, the trends in *k_t_* were more favourable than the trends in age-standardised AAMF; whereas for Lithuanian and Swedish men, and for Belgian, Finnish, and Ukrainian men and women, the recent stagnation of the increase in age-standardised AAMF was less clear or absent for *k_t_*. The cohort parameter (*g_c_*) most often evolved as an inverted U-shaped curve (Appendix A). In South-Western European countries, however, the recent cohort trends were mainly unfavourable.

We projected long-term declines in age-standardised AAMF in all 26 European countries (Figure 2). For men in Iceland, The Netherlands, and Poland, an initial increase is projected. The projected declines are stronger among men than women, which leads to convergence, except in Germany, Greece, and Italy. However, no crossovers appear.

Figure 3 compares the past and future trends in age-standardised AAMFs between the countries, and in particular between the three groups of countries that we identified. It can be observed that the projected declines are smaller for the South-Western European countries with predominantly past decelerating declines than for the North-Western European countries and Eastern European countries, which only recently experienced more rapid declines, or stagnating/ongoing increases. AAMF levels are projected to converge across countries. However, for men in particular, the projected AAMF levels in 2060 are higher in Eastern Europe and in South-Western European countries than in the North-Western European countries.

Averaged across the 26 European countries, the projected age-standardised AAMFs in 2060 are, on average, 5.1% among men, and 1.4% among women (Table 1). Among men in Western Europe, AAMF are projected to decline from, on average, 8.3% in 2016/LAY to 6.4% in 2030, 5.1% in 2045, and 4.5% in 2060. For men in Eastern Europe, AAMF are projected to decline, on average, from 14.6% in 2016/LAY to 9.5% in 2030, 7.1% in 2045, and 6.0% in 2060. Among men, the highest AAMF levels are projected for Belarus up to 2046, and for Portugal thereafter; and the lowest AAMF levels are projected in Norway up to 2053, and in Iceland thereafter. Among women, the projected AAMF levels and their decline are, on average, rather similar for Eastern and Western Europe. Iceland is expected to have the lowest AAMF levels from 2030 onwards, whereas France (up to 2057) and The Netherlands (from 2058 onwards) are expected to have the highest AAMF levels.

The projections of age-specific AAMF (see Appendix A; Appendix B
Figure A1 and Figure A2) indicate that in the majority of populations, age-specific levels will be converging. For men in Austria, Germany, Greece, Italy, The Netherlands, Slovenia, Spain, and Switzerland, for whom only moderate declines are projected, this convergence is less clearly visible. The age pattern of AAMF, which was characterised in 2016 by an inverted U-shaped curve peaking around age 50, and with high levels at younger ages as well in Western Europe, is projected to stay approximately the same in the majority of countries (See Appendix B
Figure A3 and Figure A4), albeit with some shifts in the peak age, which are particularly pronounced for populations with stagnating period declines combined with recent cohort increases (e.g., Germany (men), Greece (women), Italy, Spain, Portugal (men)).

## 4. Discussion

### 4.1. Principal Findings

Using the GBD 2017 estimates of alcohol-attributable mortality, which we adjusted at older ages, we found that, in 2016, the age-standardised AAMF were substantially higher among men (10.1%) than women (3.3%); and were much higher in Eastern Europe (14.3%) than in Western Europe (8.2%) among men. From 1990 to 2016, age-standardised AAMF mainly increased in Eastern and North-western Europe, and then declined or stagnated; whereas in South-western Europe, AAMF mostly declined, albeit with decelerations, particularly among men. Using our novel approach to project alcohol-attributable mortality, we estimated that in the future, AAMF levels will decline in all countries and will converge across countries, but that for men, levels will be higher in Eastern and South-western Europe than in North-western Europe. The projected AAMF for 2060 are, on average, 5.1% among men and 1.4% among women.

### 4.2. Evaluation of Data and Methods

We carefully assessed current levels and past trends in alcohol-attributable mortality using an estimation approach that deals with important shortcomings of previous estimates. Firstly, compared to previous research that mostly adopted an underlying cause-of-death approach [20], our estimates—which are largely based on the GBD estimates—include mortality due to alcohol from causes of death partly attributable to alcohol [22]. Consequently, our estimates are higher than those by Rosén and Haglund (2019) and by Trias-Llimós et al. (2020)—which were only based on causes of death wholly related to alcohol [18,19]—and are lower than estimates that include all deaths from causes of death partly attributable to alcohol (e.g., external causes of death), like the estimates by Pruckner et al. (2019) [17]. Secondly, we adapted the GBD alcohol-attributable mortality rates for 65+ in response to quality concerns due to the limitations of applying their estimation technique at higher ages [20,23]. Compared to the very steep increases (men) and steep declines (women) in alcohol-attributable mortality rates with age observed in the GBD data, we obtained an inverted U-shaped curve for both sexes (see Appendix A), which is considered more realistic [20]. Consequently, compared to the GBD, our estimates tend to be lower for men and higher for women (see Appendix A), and are more likely to accurately represent the age pattern of alcohol-attributable mortality, and, in turn, its cohort patterns.

The use of a certain estimation technique affects not just past alcohol-attributable mortality levels [20], but can also affect its past trends and consequently its future levels. For example, the past declines in alcohol-attributable mortality throughout Europe that Pruckner et al. [17] reported are inconsistent with our current findings and with previous findings [9,10,12] that showed different trends by country. This is likely because Pruckner et al. included mortality from all external causes [17], which are not all attributable to alcohol, and which declined throughout Europe [3].

The above illustrates that the estimation of alcohol-attributable mortality is not straightforward and could considerably affect the outcomes. Therefore, it is essential to critically assess the estimation technique before utilizing its outcomes. Despite our efforts to improve current alcohol-attributable mortality estimates, also our estimates remain estimates based on the current epidemiological evidence on the effects of alcohol on causes of death and age groups; and should be considered as such. Furthermore, we recommend investments in further improvements of estimates of alcohol-attributable mortality, particularly at older ages.

Our advanced approach to projecting alcohol-attributable mortality is, in our view, an important step forward compared to previous methodologies that mainly provided all-age estimates for the short-term future [15,16,17,18]. That is, our age-period-cohort approach enabled us to take into account important trend breaks due to changes in alcohol consumption patterns and differences between birth cohorts in alcohol-attributable mortality, and to obtain realistic future estimates of age-specific alcohol-attributable mortality. Moreover, in contrast to the previous projections by Sheron et al. and Pruckner et al. [15,16,17], which basically consisted of linear extrapolations of past trends, our projection approach is able to produce plausible long-term outcomes. That is, by transforming the outcome measure, implementing lower bounds, and assuming that increases will eventually turn into declines, we avoided not only generating long-term estimates below zero, but also projecting unlikely future crossovers and large differences in AAMF levels both between sexes and between country groups. Both kinds of outcome can easily occur when linearly extrapolating past trends for different countries with largely different past trends.

All in all, our projection provides society and health policy-makers not only with a more realistic estimate of future alcohol-attributable mortality by incorporating the experiences of other countries and different generations, but also with a more detailed outlook by providing estimates of future age-specific alcohol-attributable mortality. Thus, we have provided comparable estimates covering a longer future time horizon than previous projections. However, as is the case for any projection, our outcomes depend on the underlying assumptions.

First, our assumption that increases in AAMF will be followed by declines, which we made to avoid an unrealistic divergence in AAMF between countries, could be considered overly optimistic. Indeed, this assumption is based on the premise that there will be strong (continued) policy efforts and increased awareness of the harmful effects of alcohol in these countries (see as well Section 4.3). However, our observations of (i) more favourable cohort patterns for countries with recent period increases, and (ii) of recent declines in or the stagnation of AAMFs for selected ages in most of these countries (e.g., United Kingdom, Lithuania, The Netherlands, Poland) (Appendix A), are in line with our general assumption, which is also backed up by trends in other European countries, and by recent alcohol consumption patterns (see Section 2.4).

Second, our long-run outcomes are, logically, dependent on the lower bounds we implemented. The implementation of these lower bounds in order to avoid crossovers between the historically higher alcohol-attributable mortality levels among men than among women, and similarly, the higher alcohol-attributable mortality levels among Eastern European men than among Western European men, could be considered conservative, particularly for those countries that currently have high and strongly declining levels of alcohol-attributable mortality.

Overall, however our projections of (eventual) declines with a lower bound seem to reflect the further (decelerating) decline in alcohol consumption for Europe as a whole that was recently projected by Manthey et al. [30].

In addition, like all projections, our outcomes come with a degree of uncertainty that our projection intervals do not fully capture. First, our implementation of the lower bounds resulted in relatively small projection intervals that decrease with time. Second, more generally, projection intervals hardly ever fully reflect the full uncertainty of projection outcomes, which emerges from model uncertainty and parameter uncertainty, but also from uncertainty related to the underlying assumptions and choices [31]. In fact, we consider our projections for men in Eastern European countries more uncertain than those for other populations. For men in Ukraine and Lithuania, in particular, the combination of the assumed lower bound value with the quadratic curve extrapolation resulted in large projected declines, leading to temporal crossovers with Western European countries.

Thus, although we devised a general methodology to project alcohol-attributable mortality realistically into the long-term future, our outcomes are dependent on our assumptions. For selected countries, further methodological advancements or refinements in assumptions based on additional national knowledge would be beneficial. Moreover, for the projection of alcohol-attributable mortality in individual countries, the inclusion in the model of country-specific information on the determinants of (trends in) alcohol-attributable mortality could have added value.

In addition to the issues with the estimation of alcohol-attributable mortality and the importance of our assumptions for our projection outcomes, a third important limitation of our study is that we could not include in our analysis the foreseen, but currently unknown, exact effects of the COVID-19 pandemic. In fact, both declines due to decreases in people’s ability to afford alcohol as a result of an economic downturn and increases due to increases in hazardous drinking to cope with increased unemployment and perceived stress, can be expected [32].

### 4.3. Interpretation of Findings

We observed for the 26 European countries studied, that women have always exhibited lower AAMF levels than men. This gender gap may be related to higher alcohol consumption levels and riskier alcohol consumption patterns (binge drinking, drinking of spirits) among men than among women [33]. These sex differences in alcohol consumption are generally explained by traditional gender differences in social roles (e.g., women are responsible for performing housework, while men are responsible for generating a labour market income) [34], but also by sex differences in strategies for coping with stress, with men being more likely than women to deal with stressful situations by consuming large amounts of alcohol [35,36]. Mainly for the latter reason, we do not regard it as likely that in the future AAMF levels for men will be lower than those projected for women.

In addition, we observed higher alcohol-attributable mortality in Eastern Europe than in Western Europe, particularly among men, which also reflects differences in alcohol consumption levels and patterns [11,37]. Most Eastern European countries have a long tradition of binge drinking, of heavy consumption of vodka and other spirits, and of the homebrewing of spirits, which has resulted in high recorded and unrecorded levels of alcohol consumption, particularly among men, with very detrimental effects on health [38,39]. During the economic crisis Eastern Europe experienced in the early 1990s the risky drinking patterns among adult men in particular were aggravated [40]. Even though Eastern European countries are currently moving more and more towards beer consumption, because of their traditional patterns of heavy drinking, particularly when dealing with difficult socio-economic circumstances, we regard it as unlikely that the higher current age-standardised AAMF values in Eastern European countries for men will become lower than those in Western European countries in the future.

We observed a long-term declining trend in alcohol-attributable mortality among South-Western European countries, and recent declines or stagnating increases in most North-Western and Eastern European countries. The (decelerating) declines in Southern European countries can be linked to the move away from high levels of wine consumption, particularly during meals, and towards an increased consumption of beer, in line with wider societal changes [41]. The recent declines in alcohol consumption in Eastern Europe have been attributed to the moderate shift away from drinking spirits and towards consuming beer in a context of economic stabilisation, and were reinforced by the implementation of stricter preventive health policies from the mid-2000s onwards (including controls on the production and sale of alcohol in Russia, and increased alcohol taxes in the Baltic countries) [42,43]. For the North-Western European countries, the observed (past) increases in AAMF reflect (temporarily) increasing (Finland, Iceland, Ireland, Norway, Sweden, United Kingdom) or stagnating alcohol consumption patterns (Belgium, Denmark, The Netherlands) [7,11], which have been attributed to the increased availability and affordability of alcohol [44], combined with an expanding culture of heavy episodic drinking that is especially dangerous to health [45]. The implementation of (successful) preventive policies targeting these recent unfavourable patterns (e.g., [27]), has most likely resulted in the recent stagnation of the increase or the declines in alcohol consumption, and consequently alcohol-attributable mortality, in these countries. We projected that AAMF levels will converge across countries, but also that for men, AAMF levels will be higher in Eastern and South-western European countries than in North-western European countries. The high future levels for Eastern European men are mainly attributable to their high past levels. The high future levels for South-western European men, however, seem related to the deceleration in the decline in their alcohol-attributable mortality (Figure 1), and their more unfavourable recent cohort patterns (Appendix A). This could result from a levelling off of the decline in wine consumption [7,11] and from a recent uptake of unfavourable drinking patterns among youth [46], respectively.

In addition, we projected declines in all countries, even for selected Eastern and North-Western European populations for whom (stagnating) increasing trends have recently been observed. These projected declines are in line with our general projection approach (see Section 2.4), with recent indications in the country-specific past trends (see Section 4.2), but also rely on strong, ongoing efforts aimed at reducing excessive alcohol consumption and its negative health effects. Also, for the remainder of countries, our projections rely on the assumption that the recent favourable trends will continue. Given that these recent favourable trends are at least partly driven by effective public health efforts, continued public health action is required for these countries as well.

For selected Eastern and North-Western European countries in particular, we recommend the more extensive implementation of policies or restrictions that have proven to be effective in reducing alcohol consumption and, in turn, alcohol-attributable mortality. The aim of such policies is generally to reduce the affordability, availability and marketing of alcohol (see [27,47] for further details). For example, there is evidence that that countries that have more restrictions on alcohol advertising also have the lowest prevalence of hazardous drinking among middle-aged people [48], and that setting a minimum price for a unit of alcohol can reduce alcohol consumption among harmful drinkers [49]. It has also been shown that setting limits on the availability of alcohol (e.g., by limiting the opening hours of shops where alcohol can be sold, or by establishing age restrictions for buying alcohol) are helpful, because the density and the opening hours of outlets that sell alcohol are positively associated with alcohol-related harms [50].

However, given that previous declines in alcohol-attributable mortality seem to stem not only from the successful implementation of preventive actions, but from societal changes and shifts in the drinking culture [41], it is also important to (further) improve social awareness of the health risks of (excessive) alcohol consumption. Among the specific actions that could increase this awareness include the lowering of the norms for alcohol consumption levels in national guidelines [2], and the inclusion of health warning messages in alcohol labelling [47].

## 5. Conclusions

Our careful assessment of alcohol-attributable mortality levels and trends over time revealed important differences between European countries and men and women. Moreover, our work illustrates that the estimation of alcohol-attributable mortality is not straightforward and that investments in further improvements of estimates of alcohol-attributable mortality, particularly at older ages, would be beneficial.

Applying our novel advanced projection methodology to the past trends in 26 European countries, we were able to obtain realistic estimates of both all-age and age-specific alcohol-attributable mortality for the long-term future. We project that the share of mortality due to alcohol will decline in all countries, and will converge across countries and sexes.

However, the projected declines do not mean that no further efforts are needed to prevent excessive alcohol consumption and to counteract its negative health effects, as our projections are based on the assumption that recent favourable trends—driven in part by effective public health action—will continue and will become more widespread in Europe in line with recent indications.

For selected Eastern and North-Western European countries in particular, we recommend (further) actions to reduce the affordability, availability and advertisement of alcohol. In addition, we believe that (further) efforts to improve social awareness of the health risks of (excessive) alcohol consumption should be undertaken in order to further reduce alcohol consumption, alcohol-related health problems and, ultimately, alcohol-attributable mortality.

## Figures and Tables

**Figure 1 ijerph-17-09024-f001:**
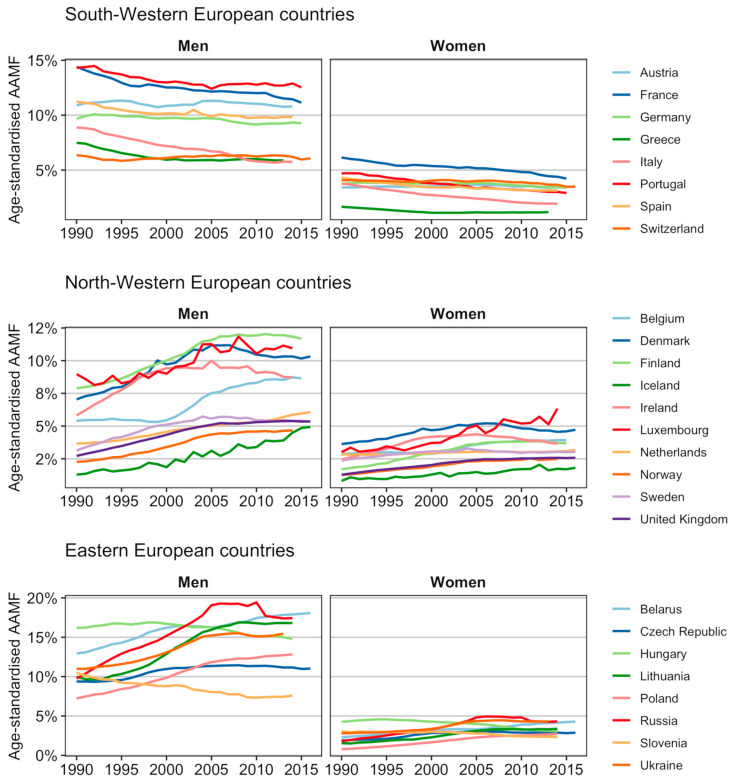
Past trends in age-standardised alcohol-attributable mortality fractions (AAMFs) (ages 20–84), 1990 up until 2016 (or the latest available year), by country and sex.

**Figure 2 ijerph-17-09024-f002:**
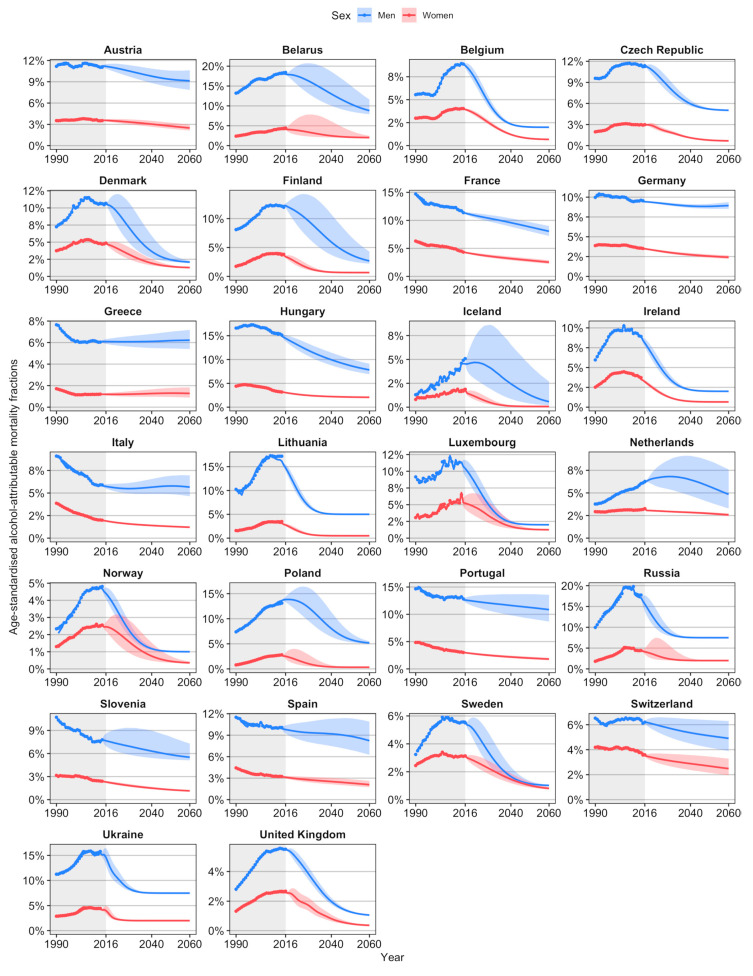
Past and future trends in age-standardised alcohol-attributable mortality fractions (ages 20–84), 1990–2060, by country and sex.

**Figure 3 ijerph-17-09024-f003:**
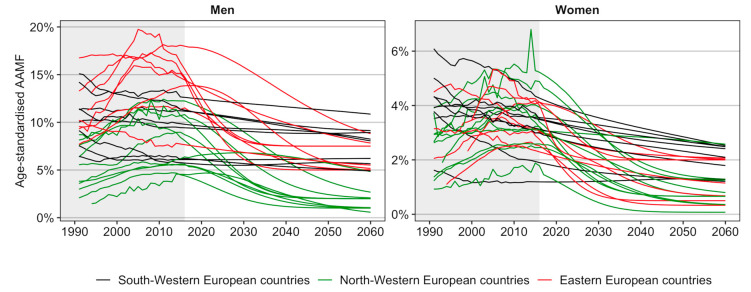
Comparison between the three groups of countries of the country-specific past (= fitted) and future (= projected) trends in age-standardised alcohol-attributable mortality fractions (AAMFs) (ages 20–84), 1990–2060, by sex.

**Table 1 ijerph-17-09024-t001:** Current and future age-standardised alcohol-attributable mortality fractions (%, ages 20–84), for selected years in 26 European countries, by country, country group (unweighted averages), and sex.

	Men				Women			
Country/Region	LAY *	2030	2045	2060	LAY	2030	2045	2060
**South-Western European Countries**								
Austria	10.79	10.40	9.53	9.13	3.43	3.28	2.89	2.48
France	11.14	10.27	9.20	8.05	4.23	3.55	3.01	2.53
Germany	9.27	9.08	8.83	8.94	3.40	2.98	2.64	2.41
Greece	5.89	6.08	6.14	6.22	1.17	1.21	1.29	1.28
Italy	5.75	5.50	5.74	5.65	1.93	1.54	1.35	1.21
Portugal	12.53	11.99	11.42	10.84	2.92	2.44	2.07	1.79
Spain	9.84	9.34	9.07	8.24	3.14	2.73	2.43	2.09
Switzerland	6.06	5.68	5.22	4.91	3.50	3.09	2.78	2.48
**North-Western European Countries**								
Belgium	8.65	4.13	2.10	2.00	3.93	2.27	0.93	0.68
Denmark	10.33	6.45	2.87	2.08	4.72	2.96	1.63	1.29
Finland	11.68	8.99	4.72	2.68	3.69	1.22	0.68	0.66
Iceland	4.93	4.02	1.81	0.58	1.81	0.43	0.09	0.08
Ireland	8.71	3.35	2.05	2.00	3.65	1.15	0.68	0.66
Luxembourg	10.95	4.34	2.11	2.00	6.34	3.04	1.46	1.26
The Netherlands	6.07	6.80	6.11	4.84	3.18	2.93	2.79	2.58
Norway	4.68	1.97	1.06	1.00	2.49	1.41	0.56	0.35
Sweden	5.34	3.06	1.32	1.02	3.03	2.12	1.21	0.80
United Kingdom	5.35	3.48	1.54	1.06	2.58	1.53	0.60	0.36
**Eastern European Countries**								
Belarus	18.07	15.77	11.69	8.81	4.29	3.21	2.25	2.02
Czech Republic	11.02	7.96	5.45	5.03	2.88	1.85	0.90	0.68
Hungary	14.78	11.54	9.29	7.83	3.08	2.41	2.16	2.07
Lithuania	16.81	6.14	5.01	5.00	3.37	0.70	0.50	0.50
Poland	12.83	11.59	6.71	5.17	2.72	0.84	0.35	0.33
Russia	17.43	8.24	7.51	7.50	4.31	2.31	2.00	2.00
Slovenia	7.61	6.84	6.07	5.52	2.33	1.77	1.38	1.15
Ukraine	15.45	8.04	7.50	7.50	4.31	2.01	2.00	2.00
**Unweighted Averages**								
Western European countries	8.22	6.39	5.05	4.51	3.29	2.22	1.62	1.39
Eastern European countries	14.25	9.51	7.40	6.54	3.41	1.89	1.44	1.34
European countries	10.08	7.35	5.77	5.14	3.32	2.11	1.56	1.37

* LAY = latest available year, which ranges from 2013 up to 2016. See Appendix B
Table A1 for the data availability by country.

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
