# Peer review of "Past and Future Alcohol-Attributable Mortality in Europe"

_ijerph, 2020, doi:10.3390/ijerph17239024_

Round 1

Reviewer 1 Report

The present manuscript studies alcohol-attributable mortality up to 2060 in 26 European. Authors used an advance forecasting age-period-cohort projection methodology which is very sound and innovative. Result mainly found important differences between European countries and men and women, that may help policymakers in taking actions regarding alcohol use and policies in different European countries.

The manuscript is well-written and structured. However, I have few comments:

  1. Although different alcohol-fatalities causes are depicted I missed a mention to alcohol-related road accidents.
  2. The rationale of a gender difference in alcohol (although historically) must be stated in the discussion section. Otherwise, an “unexplained” phenomenon is discussed along the manuscript but not disclosed the reasons of this difference.
  3. Line 99 It should be clearer for the readers why GBD estimates of alcohol- attributable mortality for the highest ages (65+) are considered implausible.
  4. It would be necessary to give a rationale of why making western, eastern and so on differences. Based on what?
  5. Some countries are not taken into account (e.g. Malta, Latvia) why?
  6. Another issue that strikes me in why authors chose 2060 as a forecasting year. Is there any reason that is based on methodological aspects o it is convenient for authors?
  7. Lines 120-121 the word “much” is quite tentative as establish “what is much” is quite arbitrary
  8. Line 315 Alcohol intake/abuse a coping strategy must be stated, and not only social or economic reasons.

Reviewer 2 Report

The article presents a description of alcohol attributable mortality in Europe up to 2060 based on the Global Burden of Disease Study applying Alcohol Attributed Mortality Fractions (AAMF). Unfortunately I found the article difficult to read and was not clear what it added to the literature.

Specifically:

1) From the methods section I found it hard to understand what analysis had been conducted. I needed to read the Kehoe et al 2012 paper to understand what the AAMF was and also how alcohol related mortality versus mortality  related to abstinence was included. I wasn't convinced regarding your statement that the AAMF reflects the additional alcohol related mortality compared to what would have occurred if the whole the whole population had been abstainers. From this statement it is hard to tell how background mortality for abstainers is included, which is much clearer in the Kohoe paper. Kohoe also state that they include abstainers AND former drinkers in the calculation, from my brief reading. Overall I feel more detail on the AAMF is required to help the reader understand the methodology employed. 

2) I also struggled to understand what the data was and how it was used in the analysis. I understood the analysis was based on the Global Burden of Disease, but from the article alone it is hard to understand what this data is (is it patient level data? How did you obtain it?)

3) I would suggest either removing or amending Figure 3. As it currently stands it is uninterpretable due to the large number of lines.

4) Overall I wasn't clear what the paper added to the current literature. From what I can figure out you are trying to show how your estimate is superior to other estimates? This is somewhat discussed in the discussion, but it is not 100% clear what the implications are. You state that your advanced approach is a step forward to current methodologies - if this is one of the key aims of the paper I feel the methods section needs to be a lot clearer to show how your methods are different (and are not just utilising those by Koebe et al). I feel the paper could more clearly state what the implications are for policy makers and the health system as a whole. My interpretation of the conclusion is currently "alcohol related mortality will decline due to public health policy, so public health policy needs to be implemented" which seems like a a weirdly circular argument. The purpose of the comparison between Eastern and Western Europe is also unclear. If the comparison is for methodological purposes then having the opening statement in the discussion being about a comparison between Eastern and Western European countries is overstating the finding in regards to the aim of the paper. If the aim of the paper is to examine Eastern versus Western drinking using a new methodology then the whole paper needs to be rewritten with this aim in mind.  

Reviewer 3 Report

The authors present an interesting work on alcohol-related mortality and trends in various countries. However, authors should make some improvements before publication:

1-You must adapt the citations and bibliographical references to the standards of the journal.

2-They must adapt the figures to the standards of the journal.

3-It is not clear to me what they attribute the downward trend and what concrete actions they propose as strategies to continue this trend.

4-The discussion should include the limitations of the study.

5-I suggest including more information and discussing more and more specifically the impact of alcohol on health.

6-Finally, they should take more account of possible unanalysed confounding variables.

One of the main strengths of this study is that it includes information from 26 countries and not only analyzes retrospectively the morbidity associated with alcohol consumption, but also, as a new aspect, it contributes to the long-term decline of alcohol consumption. The main weaknesses are minor aspects such as the adaptation of the manuscript to the standards of the journal and improvements in the discussion of alcohol-related health problems and possible confounding variables. It was also found that there is a lack of mention of limitations to be taken into consideration when interpreting the results.
